# Anti-Inflammatory and Antioxidant Activities of Medicinal Plants Used by Traditional Healers for Antiulcer Treatment

**Kanokkarn Phromnoi** [1,*]**, Puksiri Sinchaiyakij** [2]**, Chakkrit Khanaree** [3]**, Piyawan Nuntaboon** [1]**, Yupa Chanwikrai** [2]**, Thida Chaiwangsri** [4] **and Maitree Suttajit** [1]

[1]   Division of Biochemistry, School of Medical Sciences, University of Phayao, Mueang, Phayao 56000, Thailand
[2]   Division of Nutrition, School of Medical Sciences, University of Phayao, Phayao Province 56000, Thailand
[3]   School of Traditional and Alternative Medicine, Chiangrai Rajabhat University, Mueang, Chiangrai 57100, Thailand
[4]   Division of Microbiology and Parasitology, School of Medical Sciences, University of Phayao, Phayao Province 56000, Thailand
[*]   Correspondence: kanokkarn.ph@up.ac.th; Tel.: +668-9756-0120 or +66-5444-6666

**Abstract:** For centuries, many kinds of native plants and their products have been used for the treatment of gastric ulcers by traditional healers in Phayao province. The current study aimed to investigate the polyphenol content in some of these medicinal plants and to point out the relationship between their antioxidant capacity and anti-inflammatory activities. Six species were selected based on ethnopharmacologic considerations: *Punica granatum* L., *Psidium guajava* L., *Careya arborea* Roxb., *Gochnatia decora* (Kurz) Cabr., *Shorea obtusa* Wall. ex Blume, and *Ficus hispida* L.f. The leaves or bark of these plants were extracted with 70% ethanol and water. Anti-inflammatory and antioxidant activities of the extracts were analyzed based on nitric oxide (NO) and proinflammatory cytokine production in lipopolysaccharide (LPS)-stimulated RAW264.7 macrophages and through the determination of scavenging activity. The results demonstrated that the ethanol extract from *P. granatum* and *P. guajava* leaves significantly inhibited NO production by suppressing nitric oxide synthase. The extracts also inhibited tumor necrosis factor-α, interleukin-1, and interleukin-6 in terms of both mRNA and protein levels and possessed high antioxidants. These extracts were shown to contain the highest amount of polyphenols. Our study concluded that among the plants studied, *P. granatum* and *P. guajava* have the most significant anti-inflammatory and antioxidant activities and polyphenols. These plants may have the potential for use in gastric ulcer therapy due to their indicated properties. Future research should focus on the isolation of their active compounds and their in vivo biological activities. Their beneficial applications need to be warranted by such evidence.

**Keywords:** anti-inflammatory; antioxidants; medicinal plants; traditional healers; antiulcer

---

## 1. Introduction

Gastric ulcers are a common digestive disease, which are usually caused by *Helicobacter pylori* (*Hp*), non-steroidal anti-inflammatory drugs (NSAIDs), or stress. The production of proinflammatory mediators such as tumor necrosis factor-α (TNF-α), interleukin-1 (IL-1), IL-6, nitric oxide synthase (NOS), and NO is also a vital mechanism for ulceration [1–3]. Moreover, these inflammatory molecules can react with free radicals, and this can result in the development of degenerative diseases [4].

Many researchers have revealed the relevance of gastrointestinal tract hormones and gastric mucosal blood flow in the regulation of the body's physiological activities. Previous studies have shown that the administration of growth hormones can accelerate the healing of experimental

gastroduodenal ulcers [5,6]. Increasing the level of ghrelin and obestatin in a rat with gastric ulcers has also led to significant restitution of proper blood flow through mucosal microcirculation [7–10]. Moreover, the maintenance of gastric mucosal blood flow is essential for the evaluation of other experimental gastrointestinal lesions such as pancreatitis and colitis [11–15].

*Hp* infection causes an inflammation which increases the production of proinflammatory cytokines [16] and causes the stomach to produce more acid; this, in turn, leads to possible irritation and injury of the stomach lining and epithelial cells [17].

NSAIDs are widely prescribed drugs, used for the reduction of pain and inflammation; however, they can also cause gastrointestinal complications, such as ulcers and erosions [18]. NSAIDs lower the stomach's ability to make a protective layer of mucus and make it more susceptible to damage from stomach acid. NSAIDs can also affect the flow of blood to the stomach, reducing the body's ability to repair cells. The mechanism by which NSAIDs can cause mucosal injuries is a result of the inhibition of cyclooxygenase (COX) and the subsequent prostaglandin (PG) deficiencies that can occur. PG plays an essential role in gastric mucosal defense. This effect is dependent on the prostaglandin-induced stimulation of bicarbonate and mucous secretion, inhibition of gastric acid secretion, and regulation or maintenance of epithelial cell restitution and mucosal blood flow [19–21].

COX has two isoforms—COX-1 is primarily responsible for PG synthesis in the GI tract, whereas COX-2 is responsible for PG synthesis at inflammation sites. Narayanan et al. found that the patients taking COX-2 inhibitors demonstrated lower incidences of ulceration at the level of approximately 3%–5% when compared to those receiving traditional NSAIDs (nonselective; inhibiting both COX-1 and -2), which have a 20%–40% incidence rate [17], making them safer for use in the GI tract. The activities of COX-2 are necessary for the therapeutic effects of various peptides, such as growth factors, calcitonin gene-related peptides (CGRPs), as well as some gut hormones including gastrin, cholecystokinin (CCK), leptin, ghrelin, and gastrin-releasing peptides (GRPs) in the stomach. Therefore, treatment with these peptides can reverse the harmful effects of COX-1 inhibitors on the healing of ethanol-induced gastric ulcers [9,22].

Currently, several synthetic drugs are widely used for the management of inflammatory conditions; nevertheless, these drugs may be responsible for different adverse side effects such as hypersensitivity, arrhythmia, impotence, gynecomastia, and hematopoietic disorders [23]. In particular, though, they can cause gastric irritation, which can lead to the formation of gastric ulcers [24,25]. Extracts from medicinal plants, containing active compounds such as curcumin, polyphenols, flavonoids, proanthocyanidins, and tannins, have been used for ulcer treatment [26–28]. There is an abundance of medicinal plants worldwide that are used in this way, including those frequently used in Thailand to protect and heal people from ulcers [29,30]. For centuries, the consumption of picked, decocted, infused, or boiled preparations of many kinds of native plants have been used by traditional healers in Phayao province for gastric ulcer treatment. However, information about the antioxidant and anti-inflammatory effects as well as the level of polyphenols of their natural products used in the treatment of gastric ulcers has not been intensively explored and reported. Thus, this study aimed to determine the anti-inflammatory and antioxidant activities and polyphenols in six medicinal plants found in Phayao, and those potentially effective for the treatment of gastric ulcer, namely, *Punica granatum*, *Psidium guajava*, *Careya arborea*, *Gochnatia decora*, *Shorea obtusa*, and *Ficus hispida*.

## 2. Materials and Methods

### 2.1. Collection and Preparation of the Extracts

The six antiulcer medicinal plants to be studied were selected on the advice offered by Mr Kaew Wandee, a traditional healer in Baan Tham, Dok Kham Tai district, Phayao province, Thailand. They include *P. granatum*, *P. guajava*, *C. arborea*, *G. decora*, *S. obtusa*, and *F. hispida*. The findings about the potential effectiveness of these plants are similar to those found in the in vivo and in vitro studies previously reported [31–40]. Leaves from *P. granatum* and *P. guajava* and pieces of bark from

*C. arborea*, *G. decora*, *S. obtusa*, and *F. hispida* were collected in Baan Tham, Dok Kham Tai, Phayao. These botanical plants were identified by Dr. Boonchuang Boonsuk, Department of Biology, School of Science, University of Phayao. All voucher specimens were deposited at the Queen Sirikit Botanic Garden Herbarium (QBG), Mae Rim, Chiang Mai, Thailand. The voucher specimen numbers are as follows: K. Phromnoi_1 (*Punica granatum* L.); K. Phromnoi_2 (*Psidium guajava* L.); K. Phromnoi_3 (*Careya arborea* Roxb.); K. Phromnoi_4 (*Gochnatia decora* (Kurz) Cabr.); K. Phromnoi_5 (*Shorea obtusa* Wall. ex Blume); and K. Phromnoi_6 (*Ficus hispida* L.f.). The leaves or the pieces of bark from the plants were dried and finely ground. The powder was extracted using a 70% ethanol (EtOH) and water ($H_2O$) solution, with occasional shaking overnight, and they were then filtered using Whatman paper No. 1. The filtrate was removed using an evaporator and lyophilized to obtain the EtOH and water crude extracts. Each extract was stored at 20 °C and suspended in dimethyl sulfoxide (DMSO) before use.

## 2.2. Phytochemical Screening Test

The evaluation of polyphenols was performed based on the method used by Mohammadi et al [41], with some modifications. The extract was boiled in 10 mL distilled water. A few drops of 10% $FeCl_3$ solution were added. Blue-black precipitate indicated the presence of phenols. The study of tannin was performed following the standard phytochemical analysis protocol described by Broadhurst and Jones [42]. The extract was added with 1 mL of vanillin reagent, followed by one drop of HCl. The red color showed the presence of tannins. The determination of leuco-anthocyanin was performed according to the method indicated by Harborne [43], with some modifications. The extract was added in 2N HCl 2 mL and boiled for 5 min. The red color indicated leuco-anthocyanin.

## 2.3. Total Phenolic Content (TPC)

TPC was determined using the Folin–Ciocalteu method. Briefly, a 20 μL portion of the extract was mixed with 100 μL of 10% Folin- Ciocalteu reagent and 80 μL of 7.5% $Na_2CO_3$ and incubated for 30 min at room temperature. The absorbance was measured at 765 nm. TPC was estimated using a standard curve of gallic acid and expressed as milligram gallic acid equivalents per 1 g fraction (mg GAE/g fraction).

## 2.4. Total Flavonoid Content (TFC)

TFC was examined using the aluminum chloride colorimetric method. Briefly, 25 μL of the extract and 125 μL deionized water were mixed with 7.5 μL of 5% $NaNO_2$ solution. Then, a 15 μL portion of 10% $AlCl_3$ was added and incubated. Color development was performed by adding 50 μL of 1 M NaOH. The final volume of the reaction mixture was adjusted to 250 μL using deionized water. The absorbance was measured at 510 nm. TFC was calculated using a standard curve of catechin and expressed as milligram of catechin equivalents per 1 g fraction (mg CE/g fraction).

## 2.5. 2,2-Diphenyl-1-picrylhydrazyl (DPPH) Radical Scavenging Assay

The free-radical scavenging capacity of the extract was analyzed using the DPPH test according to the method used by Chumphukam et al. [44], with some modifications. Ascorbic acid and Trolox were used as a reference standard. The samples (20 μL) of various concentrations were mixed with 180 μL of freshly prepared DPPH methanolic solution and kept in the dark for 30 min; then, the absorbance at 540 nm was measured. Results were expressed as 50% DPPH decolorization ($IC_{50}$)

## 2.6. 2,2′-Azino-bis-3-ethylbenzthiazoline-6-sulfonic Acid (ABTS) Radical Scavenging Assay

The ABTS free radical-scavenging assay was performed as previously described [45], with some modifications. The ABST solution was diluted in potassium persulfate and kept in the dark for 12–14 h. Before use, this solution was diluted with distilled water to give an absorbance at 734 nm of approximately 0.70. The various concentrations of each fraction (10 μL) were mixed with 990 μL of working diluted ABTS and incubated for 6 min in the dark. The decrease in absorbance was measured

at 734 nm. The reference standards were Trolox and ascorbic acid. Results were expressed as 50% ABTS decolorization ($IC_{50}$).

### 2.7. Cell Culture

Mouse macrophage RAW 264.7 cells (ATCC Manassas, VA, USA) were cultured in DMEM containing 10% heat-inactivated fetal bovine serum (FBS), 100 U/mL penicillin-streptomycin under 5% $CO_2$, and 95% air at 37 °C.

### 2.8. Cell Viability Assay

The RAW 264.7 cells ($5 \times 10^3$ cells/100 mL/well) were seeded into each well of a 96-well plate and incubated for 24 h. Then, the extracts were treated at different concentrations (0-800 µg/mL) for 24 h. A 20 µL portion of 3-(4,5-dimethylthiazol-2yl)-2,5-diphenyltetrazolium bromide (MTT) was added and further incubated at 37 °C for 4 h. The excess MTT dye solution was removed, and only MTT formazan that stained the living cells was redissolved in DMSO. The color intensity was measured at 540 and 630 nm using a microplate reader [46].

### 2.9. NO Production Assay

The NO production was determined using Griess colorimetric assay, according to the manufacturer's protocol (Sigma-Aldrich, St. Louis, MO, USA) and modified from Chumphukam et al. [44]. The RAW 264.7 cells ($5 \times 10^3$ cells/100 mL/well) were incubated with a non-toxic dose of the extract ((0-800 µg/mL) for 2 h. Then, 1 µg/mL LPS was added to induce inflammation, and further incubated for 24 h. Afterward, the culture medium was mixed with 100 µL of Griess reagent and incubated at room temperature for 15 min before measuring the color change at 540 nm using a microplate reader.

### 2.10. Total RNA Extraction and cDNA Preparation

Total RNA of RAW 264.7-treated cells was isolated using a NucleoSpin®RNA kit (Macherey-Nagel, Düren, Germany). RNA was dissolved with RNase-free water; we then measured the RNA concentration and synthesized the complementary DNA (cDNA) using a ReverTra Ace®qPCR RT Kit (TOYOBO, Osaka, Japan).

### 2.11. Quantitative Real-Time PCR (qPCR)

To measure iNOS, COX-2, TNF-α, IL-6, and IL-1β mRNA expression, the cDNA was amplified in a 7500 Real-time PCR system (Applied Biosystem, Thermo Fisher Scientific, Waltham, Massachusetts, USA) using a SensiFAST SYBR®Lo-ROX Kit (Bioline, Singapore) as described in the manufacturer's protocol. GAPDH was used as a reference gene [47].

### 2.12. Enzyme-Linked Immunosorbent Assay (ELISA)

The LPS-induced TNF-α, IL-6, and IL-1β secretion in RAW 264.7 cells was analyzed using a sandwich ELISA assay kit (Biolegend, San Diego, CA, USA). Culture medium was collected and detected the production of proinflammatory cytokines according to the manufacturer's protocol.

### 2.13. Statistical Analysis

The statistical analysis was determined using one-way ANOVA. The significant differences at the levels of $p < 0.05$, $p < 0.01$ and $p < 0.001$ were determined by Tukey's honestly significant difference multiple comparison test using IBM SPSS Statistics 22 (IBM Corp., Armonk, NY, USA). Excel software was used to plot the graphs.

## 3. Results and Discussion

### 3.1. Phytochemical Screening

In general, plants that are classified as medicinal can contain many groups of phytochemicals, mainly polyphenols, tannins, and leuco-anthocyanins, which have pharmacological properties (e.g., antiulcer, antioxidant, and anti-inflammatory activities, and others) [48–52]. For this report, polyphenols and classes of polyphenols, including tannins and leuco-anthocyanins, were colorimetrically measured in a sample of six selected medicinal plants found in Phayao (Table 1). The results indicated that each extract contained different ingredients, which could lead to various % yields, colors, and differences in appearance for the extracts [53,54].

**Table 1.** Phytochemical screening test.

| Scientific Name | Used Part | Polyphenols | Tannins | Leuco-Anthocyanins |
|---|---|---|---|---|
| *Punica granatum* L. | Leaf | † | † | − |
| *Psidium guajava* L. | Leaf | † | † | − |
| *Careya arborea* Roxb. | Bark | † | † | † |
| *Gochnatia decora* (Kurz) Cabr. | Bark | † | − | − |
| *Shorea obtusa* Wall. ex Blume | Bark | † | † | − |
| *Ficus hispida* L.f. | Bark | † | − | † |

† signifies found; − signifies not found.

### 3.2. TPC and TFC of the Extracts

TPC and TFC results found that the highest amount of phenolics was detectable in *P. granatum* EtOH extracts, and the lowest amount in *F. hispida* water extracts. The EtOH extracts of *P. guajava* contained the highest amount of flavonoids, whereas the lowest amounts were showed in *S. obtusa* EtOH extracts (Table 2). This result is similar to those of previous reports that have found *P. granatum* and *P. guajava* EtOH extracts to have high levels of polyphenols and flavonoids [31,55,56]. The most active compound in *P. guajava* was found to be quercetin along with two flavonoid compounds, namely, quercetin-3-O-glucopyranoside and morin [57]. Most of the EtOH extracts showed higher phenolic and flavonoid contents than the water extracts. Interestingly, traditional healers prepare these antiulcer plants by pickling them in alcohol more frequently than by boiling them in water or through decoction. Therefore, in our study, only EtOH plant extracts were selected for further experimentation.

**Table 2.** Total phenolic and flavonoid compounds contained in the EtOH and water extracts.

| Medicinal Herbs | TPC (mg GAE/g Extract) | | TFC (mg CAE/g Extract) | |
|---|---|---|---|---|
| | EtOH | Water | EtOH | Water |
| *P. granatum*/leaf | 410.04 ± 2.06 [e] | 235.79 ± 0.89 [c] | 23.39 ± 2.10 [b] | 13.06 ± 1.74 [a] |
| *P. guajava*/leaf | 315.77 ± 1.41 [d] | 274.78 ± 9.51 [d] | 90.83 ± 1.15 [d] | 59.58 ± 3.56 [c] |
| *C. arborea*/bark | 252.34 ± 11.88 [c] | 208.58 ± 10.36 [c] | 27.32 ± 1.52 [b] | 34.54 ± 5.51 [b] |
| *G. decora*/bark | 62.99 ± 4.89 [a] | 98.64 ± 7.17 [b] | 28.24 ± 3.2 [b] | 25.68 ± 3.08 [b] |
| *S. obtusa*/bark | 332.15 ± 24.01 [d] | 301.81 ± 21.36 [d] | 4.54 ± 1.32 [a] | 8.25 ± 1.39 [a] |
| *F. hispida*/bark | 151.49 ± 5.70 [b] | 58.91 ± 4.13 [a] | 77.45 ± 4.51 [c] | 26.28 ± 0.80 [b] |

The values are expressed as mean ± SD ($n = 3$). Means with different letters in the same column are significantly different ($p < 0.05$).

### 3.3. DPPH and ABTS Radical Scavenging Assay

In this assay, the antioxidant capacity of the EtOH extracts was determined through the inhibition of DPPH and ABTS radicals. For the DPPH assay, the highest antioxidant activity ($IC_{50}$) was shown in *P. granatum*, followed by *C. arborea*, *P. guajava*, *S. obtusa*, *F. hispida*, and *G. decora*, respectively. For the ABTS assay, *P. granatum* was also shown to have the highest antioxidant activity, followed by *C. arborea*,

*S. obtuse*, *P. guajava*, *F. hispida*, and *G. decora*, respectively. Notably, our results demonstrated that the $IC_{50}$ values of *P. granatum* and *P. guajava* were nearly equal to the values of ascorbic acid and Trolox, used as a reference standard (Table 3). In a comparison of the antioxidant capacity determined by ABTS and DPPH assays, our results showed that the activity recorded by the former test was significantly higher than in the latter one. Furthermore, ABTS is used to target oxygen radicals commonly found in our bodies, whereas DPPH is used to target nitrogen radicals that are rarely presented in living organisms [58,59]. These studies confirm that the ABTS assay is more reliable than the DPPH assay for determining antioxidant activity in numerous natural products. Our results are similar to those of the previous studies, reported by Bekir et al., that the methanolic extract from *P. granatum* leaves presented an excellent $IC_{50}$ using DPPH and ABTS assays [60]. Inconsistent with the study of Amjad and Shafighi, it was suggested that the antioxidant activity of *P. granatum* leaves had a direct relationship with phenolic compounds [61]. Fernandes et al. also found that *P. guajava* leaf extracts had significant antioxidant and anti-inflammatory activities in vitro and in human cells [62]. With respect to *C. arborea*, the study of Senthilkumar et al. confirmed that *C. arborea* bark aqueous and methanol extracts contained high total phenolic content and demonstrated potent antioxidant activity against many oxidants in vitro [63].

**Table 3.** Antioxidant activity of the EtOH extracts by DPPH and ABTS methods.

| Medicinal Herbs | $IC_{50}$ (µg/ml) | |
| --- | --- | --- |
| | DPPH Assay | ABTS Assay |
| *P. granatum*/leaf | 8.72 ± 0.64 [b] | 2.21 ± 0.01 [a] |
| *P. guajava*/leaf | 11.62 ± 0.49 [b] | 3.77 ± 0.16 [c] |
| *C. arborea*/bark | 10.15 ± 0.05 [b] | 2.75 ± 0.08 [b] |
| *G. decora*/bark | 94.21 ± 3.12 [e] | 20.52 ± 0.56 [e] |
| *S. obtusa*/bark | 19.75 ± 0.45 [c] | 2.85 ± 0.03 [b] |
| *F. hispida*/bark | 46.88 ± 0.82 [d] | 10.00 ± 0.04 [d] |
| Ascorbic acid | 6.81 ± 0.01 [a] | 2.44 ± 0.10 [a] |
| Trolox | 8.83 ± 0.07 [b] | 3.42 ± 0.01 [c] |

The values are expressed as mean ± SD (*n* = 3). Means with different letters in the same column are significantly different ($p < 0.05$).

### 3.4. Effects of the Extracts on Cell Viability

The potential cytotoxicity of the extracts was evaluated using MTT assay, after incubating the cells for 24 h with 0-800 µg/mL extract. All crude extracts did not show a cytotoxic effect at a concentration of up to 800 µg/mL extract, compared to that of non-treated control. Thus, the anti-inflammatory effect of a non-toxic dose (0-800 µg/mL) was subsequently determined.

### 3.5. Effects of the Extracts on the Production of NO

NO, a well-known proinflammatory mediator, is synthesized by iNOS and is involved in many physiological and pathological process. As a result, the suppression of NO production has been characterized as an effective new pharmacological strategy for the treatment of inflammation-related disease [64]. As displayed in Figure 1, only the extracts from *P. granatum* and *P. guajava* leaves were shown to markedly inhibited LPS-induced NO production as compared with the LPS-treated group. These results are confirmed by a previous report that showed that *P. guajava* leaf extract significantly inhibited the production of NO and prostaglandin E2 (PGE2) [65]. According to Berkoz and Allahverdiyev's experiments, punicalagin, hydrolysable tannins, isolated from *P. granatum* can decrease the production of NO in a dose-dependent manner without affecting the viability of cells [66]. *P. granatum* and *P. guajava* leaf extracts exhibited high antioxidant activity and a high level of No inhibition, but the bark extracts of *C. arborea*, *G. decora*, *S. obtusa*, and *F. hispida* did not show such high levels of these activities. Therefore, only *P. granatum* and *P. guajava* extracts were further

evaluated in terms of the anti-inflammatory effects of LPS-stimulated RAW 264.7 macrophages. Indeed, the researchers sought to identify the mechanisms responsible for these effects.

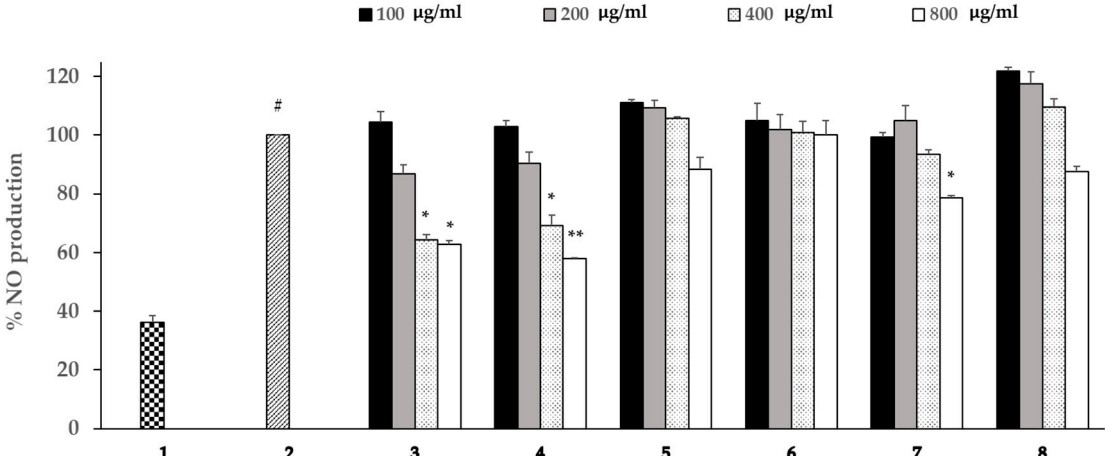

**Figure 1.** Effects of the extracts on the production of NO in LPS-stimulated RAW 264.7 macrophages. Cells were pretreated with various concentrations (0–800 µg/mL) of each extract for 1 h and then stimulated with LPS (1 µg/mL) for 18 h. NO production was determined using Griess reagent. 1, control; 2, LPS treatment; 3, *P. granatum*; 4, *P. guajava*; 5, *C. arborea*; 6, *G. decora*; 7, *S. obtusa*; 8, *F. hispida*. The data illustrates the mean of three independent experiments, each performed in triplicate (n=3). Error bars indicate SD. (Significant versus the non-treated control, # $p < 0.001$; significant versus LPS treatment, * $p < 0.05$, ** $p < 0.01$).

### 3.6. Effects of the Extracts on Proinflammatory Cytokine Expression and Production

Upon an occurrence of inflammation in the body, macrophages are activated to produce proinflammatory mediators (NO), which are synthesized by iNOS. Other proinflammatory cytokines, TNF-α, IL-β, and IL-6, are known to contribution to tissue damage and multiple organ failure. They are also considered to be important initiators of the inflammatory response and mediators of the development of various inflammatory diseases [67]. Moreover, COX-2 is an enzyme that generates PG, which is induced by proinflammatory cytokines and other activators, such as LPS, resulting in the release of a large amount of PGE2 in inflammation sites [68]. Therefore, the identification of COX-2 inhibitors is considered to be a promising approach for protecting against inflammation and tumorigenesis.

The result of the qPCR assay (Figure 2) found that the leaf extracts ameliorated inflammation through the inhibition of the inflammatory mediators; see the iNOS, TNF-α, IL-6, and IL-1β mRNA expression below (Figure 2A–D). However, the extracts could not inhibit LPS-stimulated COX-2 mRNA expression (Figure 2E). For the ELISA assay (Figure 3), the extracts also decreased the subsequent release of cytokines, namely, TNF-α, IL-6, and IL-1β. This is inconsistent with a previous study that demonstrated the promising acute anti-inflammatory activity of *P. guajava* leaf in Wistar rats [69]. *P. guajava* leaf was also shown to have suppressed the expression and activity of both iNOS and COX-2 through the downregulation of ERK1/2 activation [65]. *P. granatum* leaf extracts can be used to reduce NO production and cytokine gene expression during LPS treatment [70]. Hydrolyzable tannins from *P. granatum* were also shown to have significantly decreased carrageenan-induced mice paw edema and to have inhibited iNOS and COX-2 expression [66,71].

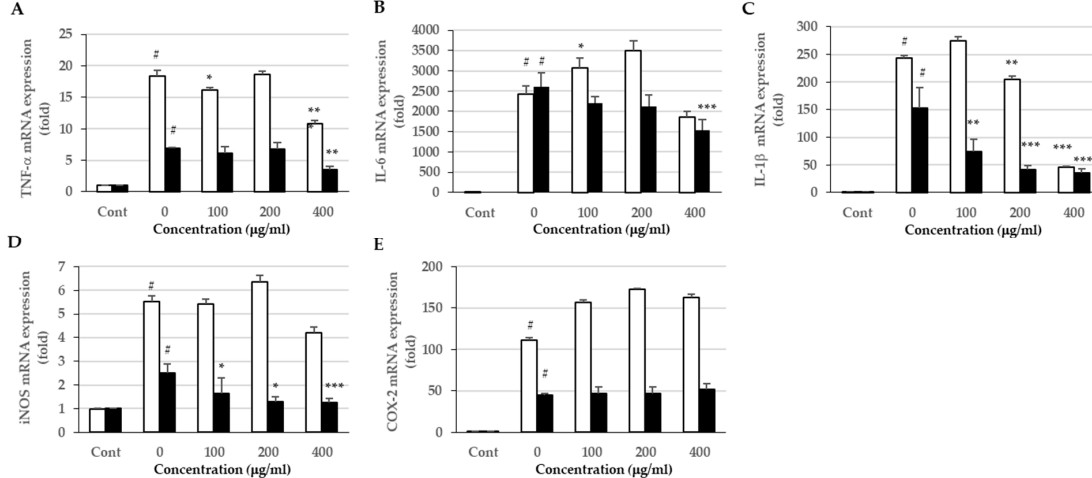

**Figure 2.** Effects of the extracts on (**A**) TNF-α, (**B**) IL-6, (**C**) IL-1β, (**D**) iNOS, and (**E**) COX-2 expression in LPS-stimulated RAW 264.7 macrophages. Total RNA was extracted and analyzed for mRNA expression. White bar represents *P. granatum*, black bar represents *P. Guajava*. (Significant versus the non-treated control, # $p < 0.001$, Significant versus LPS treatment, * $p < 0.05$, ** $p < 0.01$ and *** $p < 0.001$).

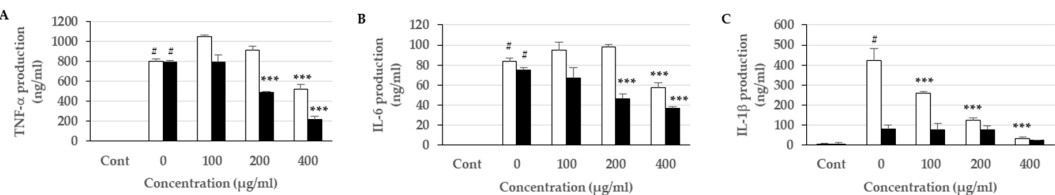

**Figure 3.** Effects of the extracts on (**A**) TNF-α, (**B**) IL-6, (**C**) IL-1β production in LPS-stimulated RAW 264.7 macrophages. Cytokines present in the culture media were measured using ELISA. White bar represents *P. granatum*, black bar represents *P. Guajava* (Significant versus the non-treated control, # $p < 0.001$, Significant versus LPS treatment, * $p < 0.05$, ** $p < 0.01$ and *** $p < 0.001$).

## 4. Conclusions

In conclusion, six medicinal plants were selected according to the advice offered by the healer, with respect to their potential effectiveness in antiulcer treatment. Our results showed that only the leaves of *P. granatum* and *P. guajava* have powerful radical scavenging capacities as well as strong anti-inflammatory activities. The leaf extracts were shown to have ameliorated inflammation through inhibition of both mRNA and protein levels of inflammatory mediators. It is proposed that such activities may be caused by the presence of a high amount of different polyphenols. The findings support the effectiveness of medicinal plants used by traditional healers in the treatment of gastric ulcers. Future research should verify the effects of *P. granatum* and *P. guajava* leaf extracts in experimentally induced ulcers, and point out the bioactive phytochemicals.

**Author Contributions:** Conceptualization, P.S., K.P. and M.S.; Methodology, K.P., P.S., P.N., and C.K.; Project Administration, T.C.; Supervision, M.S.; Writing—original draft, K.P. and Y.C.; Writing—review and editing, K.P. and M.S.

**Funding:** This research was funded by the University of Phayao Research Fund, grant number RD59051 and RD62053 and the Thailand Research Fund (TRF), grant number MRG5980170.

**Acknowledgments:** The authors wish to express their gratitude to the University of Phayao and TRF for providing supporting funds. Thanks to Kaew Wandee, a native herbal healer of Baan Tham, Dok Kham Tai district, Phayao province, for providing the plant materials and information, and Boonchuang Boonsuk, who helped identify the plants. The authors are grateful for the facilities and technical support from the School of Medical Sciences, University of Phayao, and Pornngarm (Limtrakul) Dejkriengkraikul, Department of Biochemistry, Faculty of Medicine, Chiang Mai University.

**Conflicts of Interest:** The authors declare no conflict of interest.

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
