# Peer review of "Anti-Inflammatory and Antioxidant Activities of Medicinal Plants Used by Traditional Healers for Antiulcer Treatment"

_scipharm, doi:10.3390/scipharm87030022_

Round 1
Reviewer 1 Report
The subject of the research presented in this article is very interesting. The search for new agents and substances to prevent and treat stomach damage is a topic of interest for many research teams. Topics selected by the authors are included in the subject matter of these studies. Authors in their manuscript repeatedly cited articles from leading research centers, including those recently published in leading scientific journals. After reading this article submitted to me for review, however, it occurred to me observations and comments. The described comments and suggested changes in the text lead to a better understanding of the theme and will increase readers' interest in this topic. Here they are.
1. The introduction section is too short. It is necessary to refer to the other articles. In the Helicobacter pylori (Hp) infection is involved in various gastroduodenal pathologies. Hp infection increases production of pro-inflammatory IL-1beta (PMID: 12138227). There are review articles presenting a history of the gastrointestinal tract endocrinology, as well as a relevance of gastrointestinal tract hormones in the regulation of body physiological activity.(PMID: 25716961).
2. The maintenance of normal blood flow through microcirculation plays a fundamental role in the protection and healing of mucosa in the (PMID: 1505283). In published previous studies, gastric blood flow was one of many, but very important parameters evaluated in the various experimental model of gastric ulcers. For example – PMID: 21673369, PMID: 19439811. This parameter- gastric mucosal blood flow - is also very important in the evaluation of other experimental gastrointestinal lesions (for example pancreatitis and colitis PMID: 26713317, 26769837, 25912801, 25594510, 26798415…). Administration of various factors during gastric ulcers led to a significant restitution of appropriate blood flow through mucosal microcirculation. The authors have not written about it. This should be added and discussed.
3. Prostaglandins play an essential role in gastric mucosal defense. This effect is dependent on the prostaglandin-induced stimulation of bicarbonate and mucous secretion, inhibition of gastric acid secretion, and regulation of maintaining epithelial cell restitution and mucosal blood flow (PMID: 21373261). Activity of COX-2 is necessary for the therapeutic effects various peptides in the stomach whereas treatment with these peptides can reverse a deleterious effect of COX-1 inhibitor on healing of ethanol-induced gastric ulcers (PMID: 24622834). This should be discussed.
Such a short extension of this topic will undoubtedly raise the quality of this manuscript.
Author Response
Dear Reviewer 1,
Response to Reviewer 1 Comments
Point 1: The introduction section is too short. It is necessary to refer to the other articles. In the Helicobacter pylori (Hp) infection is involved in various gastroduodenal pathologies. Hp infection increases production of pro-inflammatory IL-1beta (PMID: 12138227). There are review articles presenting a history of the gastrointestinal tract endocrinology, as well as a relevance of gastrointestinal tract hormones in the regulation of body physiological activity. (PMID: 25716961).
Response 1:
In the introduction section, Hp infection and gastrointestinal tract hormones which involved in gastroduodenal pathologies and physiologies had already been added as shown at line…52.. ref 16 & 17 (PMID: 12138227, PMID: 30228726) and line…45…ref 5 & 6 (PMID: 25716961, PMID: 15277985)
“Hp infection causes an inflammation which increases the production of pro-inflammatory cytokines [16] and causes the stomach to produce more acid; this, in turn, leads to possible irritation and injury of the stomach lining and epithelial cells [17].”
“Many researchers have revealed the relevance of gastrointestinal tract hormones and gastric mucosal blood flow in the regulation of the body’s physiological activities. Previous studies have shown that the administration of growth hormones can accelerate the healing of experimental gastroduodenal ulcers [5,6].”
Point 2: The maintenance of normal blood flow through microcirculation plays a fundamental role in the protection and healing of mucosa in the (PMID: 1505283). In published previous studies, gastric blood flow was one of many, but very important parameters evaluated in the various experimental model of gastric ulcers. For example – PMID: 21673369, PMID: 19439811. This parameter- gastric mucosal blood flow - is also very important in the evaluation of other experimental gastrointestinal lesions (for example pancreatitis and colitis PMID: 26713317, 26769837, 25912801, 25594510, 26798415…). Administration of various factors during gastric ulcers led to a significant restitution of appropriate blood flow through mucosal microcirculation. The authors have not written about it. This should be added and discussed.
Response 2:
The information about gastric mucosal blood flow had already been added
and discussed as presented in line…45…ref 7-10 (PMID: 19439811, 1505283, 24622834, 21673369) and ref 11-15 (PMID: 26713317, 26769837, 25912801, 25594510, 26798415
“Many researchers have revealed the relevance of gastrointestinal tract hormones and gastric mucosal blood flow in the regulation of the body’s physiological activities.”
“Increasing the level of ghrelin and obestatin in a rat with gastric ulcers has also led to significant restitution of proper blood flow through mucosal microcirculation [7-10]. Also, the maintenance of gastric mucosal blood flow is essential for the evaluation of other experimental gastrointestinal lesions such as pancreatitis and colitis [11-15].”
Point 3: Prostaglandins play an essential role in gastric mucosal defences. This effect is dependent on the prostaglandin-induced stimulation of bicarbonate and mucous secretion, inhibition of gastric acid secretion, and regulation of maintaining epithelial cell restitution and mucosal blood flow (PMID: 21373261). Activity of COX-2 is necessary for the therapeutic effects various peptides in the stomach whereas treatment with these peptides can reverse a deleterious effect of COX-1 inhibitor on healing of ethanol-induced gastric ulcers (PMID: 24622834). This should be discussed.
Response 3:
NSAIDs, prostaglandins, including COX-1, and COX-2 which mainly involved in gastric ulcers had already been informed and discussed as shown in the introduction part line…55.. ref 9, 17-22 (PMID: 24622834, 30228726, 19240698, 28242110, 21373261, 10792129, 16247188)
“NSAIDs are widely prescribed drugs, use for the reduction of pain and inflammation; however, they can also cause gastrointestinal complications, such as ulcers and erosions [18]. NSAIDs lower the stomach's ability to make a protective layer of mucus and make it more susceptible to damage from stomach acid. NSAIDs can also affect the flow of blood to the stomach, reducing the body's ability to repair cells. The mechanism by which NSAIDs can cause mucosal injuries is a result of the inhibition of cyclooxygenase (COX) and the subsequent prostaglandins (PGs) deficiencies that can occur. PGs play an essential role in gastric mucosal defense. This effect is dependent on the prostaglandin-induced stimulation of bicarbonate and mucous secretion, inhibition of gastric acid secretion, and regulation or maintenance of epithelial cell restitution and mucosal blood flow [19-21].”
“Cox has two isoforms, COX-1 is primarily responsible for PG synthesis in the GI tract, whereas COX-2 is responsible for PG synthesis at sites of inflammation. Narayanan et al. found that the patients taking COX-2 inhibitors have demonstrated lower incidences of ulceration at the level of approximately 3–5% when compared to those receiving traditional NSAIDs (nonselective; inhibiting both COX-1 & -2), which have a 20–40% incidence rate [17], making them safer for use in the GI tract. The activities of COX-2 are necessary for the therapeutic effects of various peptides, such as growth factors, calcitonin gene-related peptides (CGRP), as well as some gut hormones including gastrin, cholecystokinin (CCK), leptin, ghrelin and gastrin-releasing peptides (GRP) in the stomach. Therefore, treatment with these peptides can reverse the harmful effects of COX-1 inhibitors on the healing of ethanol-induced gastric ulcers [9,22].”
Reviewer 2 Report
The article reports the investigation of several plants used in the traditional Thai medicine for the treatment of gastric ulcer, with regard to their content in polyphenols, antioxidant (DPPH, ABTS assays), and anti-inflammatory effects. Some shortcomings were noted.
1. The Abstract of the paper contains some erroneous expressions and statements:
Ø Although oxidative damage and inflammation are common denominators in the pathogeny of may diseases, it cannot be stated that “Gastric ulcer is usually caused by the lack of anti-oxidation and anti-inflammation”
Ø gastritis and gastric ulcer cannot by considered synonyms
Ø Line 20: “their anti-inflammatory and antioxidant activities have never been explored and reported”. This statement is misleading, as there are several previous publications which have investigated the anti-inflammatory and antioxidant activities of Punica granatum, Psidium guajava, Careya arborea, Ficus racemosa. Only Shorea obtusa has not been researched in this reagard (as far as the reviewer knows, based on scientific articles written in English)
Ø line 23: the plants listed in this line include mostly trees, as such the term “herb” should be replaced with plants. Throughout the entire article do not refer to trees as being herbs.
Ø The exact nature of the plant parts extracted with 70% ethanol and water should be specified.
Ø line 24: use anti-inflammatory activity instead of anti-inflammation
2. The introduction should offer more relevant information on the topic, and some erroneous statements have to be corected:
Ø line 54: The antiulcer effectiveness of a plant extract cannot be considered the sum of the antioxidant and anti-inflammatory effects, nor is it the general consequence of polyphenol content. The submitted article may state that it investigates the antioxidant and anti-inflammatory effects of natural products used in the treatment of gastric ulcer, but since it doesn’t evaluate the anti-ulcer effects in any model of experimental gastric ulcer, the current article should not state that it investigates the antiulcer effectiveness of Thai plants.
3. The Materials and Methods section provides insufficient information:
Ø The botanical identity of any plant part is crucial in researches of biological activity. The section “2.1 Collection and preparation of extracts” has to mention exactly which part has been collected for each species. Furthermore, the identification of these plants by a certified botanist is needed.
Ø A voucher specimen for each plant has to be deposited in an official Herbarium and voucher codes have to be mentioned in the article. Once in the article the complete botanical name of the species (including the name of the botanist) should be given, for example: Punica granatum L., Psidium guajava L., Morinda coreia Buch.-Ham., etc.
Ø Line 68: “A polyphenol was performed...” should be replaced with “the evaluation of polyphenols was performed...”; The same applies to “A leuco-anthocyanin was determined..” in line 73
4. The Results and Discussion section needs improvement
Ø Table 1: The absence of polyphenols from several plants is not credible, as these compounds are ubiquitous in the Plant Kingdom.
Ø Since the selection of the 6 species researched in detail relied on the recommendation of the healer and not the screening reported in table 1, the reviewer considers that the inclusion of table 1 in the article is questionable.
Ø Line 199: “Cell Viability of the extracts” should be replaced with “the effects of extracts on cell viability”
5. A thorough check of the English grammar is necessary.
Ø Line 263 “All the findings can explain the antiulcer therapeutic” replace with “The findings support the effectiveness in the treatment of gastric ulcer”
Ø Line 265 “herbal plants” replace with “plants” or with “herbal products”
Author Response
Dear Reviewer 2,
We had already been responded to your comments point by point.
Please see the attachment.

Round 2
Reviewer 1 Report
The authors have completed recommendations. In my opinion, this article after the changes is good.
Author Response
Dear Reviewer,
Thank you very much for your comments and suggestions.
Best Regards,
Dr. Kanokkarn Phromnoi
Reviewer 2 Report
Significant improvements were performed by the authors in oder to give the revised version. Some minor issues are to be dealt with before publication:
- Abstract: Please remove first sentence “Gastric ulcers are usually caused by Helicobacter pylori, NSAIDs, and stress” and start the abstract with the second sentence.
- Change text from lines 22-26 into: “The current study aimed to investigate the polyphenol content in some of these medicinal plants and to point out the relationship between their antioxidant capacity and anti-inflammatory activities. Six species were selected based on ethnopharmacologic considerations: Punica granatum L., Psidium guajava L., Careya arborea Roxb., Gochnatia decora (Kurz) Cabr., Shorea obtusa Wall. ex Blume, and Ficus hispida L.f.
- Line 86: remove Aloe vera as it is not a class of secondary metabolites like the other ones in the sentence.
- Change text from lines 339-340 into: “Future research should verify the effects of Punica granatum and Psidium guajava leaf extracts in experimentally induced ulcers, and point out the bioactive phytochemicals”
Author Response
Dear Reviewer,
Response to Reviewer 2 Comments
Point 1: Abstract: Please remove first sentence “Gastric ulcers are usually caused by Helicobacter pylori, NSAIDs, and stress” and start the abstract with the second sentence.
Response 1: “Gastric ulcers are usually caused by Helicobacter pylori, NSAIDs, and stress” had already been removed as shown at line…18.
Point 2: Change text from lines 22-26 into: “The current study aimed to investigate the polyphenol content in some of these medicinal plants and to point out the relationship between their antioxidant capacity and anti-inflammatory activities. Six species were selected based on ethnopharmacologic considerations: Punica granatum L., Psidium guajava L., Careya arborea Roxb., Gochnatia decora (Kurz) Cabr., Shorea obtusa Wall. ex Blume, and Ficus hispida L.f.
Response 2: Text from line 22-26 had already been changed into “The current study aimed to investigate the polyphenol content in some of these medicinal plants and to point out the relationship between their antioxidant capacity and anti-inflammatory activities. Six species were selected based on ethnopharmacologic considerations: Punica granatum L., Psidium guajava L., Careya arborea Roxb., Gochnatia decora (Kurz) Cabr., Shorea obtusa Wall. ex Blume, and Ficus hispida L.f.” as presented at line…19-23.
Point 3: Line 86: remove Aloe vera as it is not a class of secondary metabolites like the other ones in the sentence.
Response 3: “Aloe vera” had already been removed as shown at line…79.
Point 4: Change text from lines 339-340 into: “Future research should verify the effects of Punica granatum and Psidium guajava leaf extracts in experimentally induced ulcers, and point out the bioactive phytochemicals”
Response 4: Text from line 339-340 had already been changed into “Future research should verify the effects of Punica granatum and Psidium guajava leaf extracts in experimentally induced ulcers, and point out the bioactive phytochemicals” as presented at line…297-298.
Best Regards,
Dr. Kanokkarn Phromnoi